# A Novel Groundwater Burial Depth Prediction Model Based on Two-Stage Modal Decomposition and Deep Learning

**DOI:** 10.3390/ijerph20010345

**Published:** 2022-12-26

**Authors:** Xianqi Zhang, Zhiwen Zheng

**Affiliations:** 1Water Conservancy College, North China University of Water Resources and Electric Power, Zhengzhou 450046, China; 2Collaborative Innovation Center of Water Resources Efficient Utilization and Protection Engineering, Zhengzhou 450046, China; 3Technology Research Center of Water Conservancy and Marine Traffic Engineering, Zhengzhou 450046, China

**Keywords:** groundwater burial depth prediction, complete ensemble empirical mode decomposition with adaptive noise (CEEMDAN), variational modal decomposition (VMD), convolutional neural network (CNN), gated recurrent unit (GRU)

## Abstract

The variability of groundwater burial depths is critical to regional water management. In order to reduce the impact of high-frequency eigenmodal functions (IMF) generated by complete ensemble empirical mode decomposition with adaptive noise (CEEMDAN) on the prediction results, variational modal decomposition (VMD) is performed on the high frequency IMF components after the primary modal decomposition. A convolutional neural network-gated recurrent unit prediction model (CNN-GRU) is proposed to address the shortcomings of traditional machine learning which cannot handle correlation information and temporal correlation between time series. The CNN-GRU model can extract the implicit features of the coupling relationship between groundwater burial depth and time series and further predict the groundwater burial depth time series. By comparing the prediction results with GRU, CEEMDAN-GRU, and CEEMDAN-CNN-GRU models, we found that the CEEMDAN-VMD-CNN-GRU prediction model outperformed the other prediction models, with a prediction accuracy of 94.29%, good prediction results, and high model confidence.

## 1. Introduction

The variation in groundwater depth of burial is influenced by a number of factors, including rainfall, evaporation, and the amount of extraction. Groundwater leakage and ground subsidence occur when groundwater is over-exploited. When recharge exceeds extraction, groundwater depths become shallower [1,2]. The People’s Victory Canal Irrigation District, an important grain-producing area in Henan Province, promotes the sustainable development of Chinese agriculture and provides security for China’s food security. Therefore, accurate prediction of changes in groundwater depth in the irrigation area of the People’s Victory Canal is essential for water resource management in the irrigation area [3]. Groundwater burial depth variation is a complex system, showing non-linearity, non-stationarity, local fluctuations, and multiple time scales, which makes groundwater burial depth prediction more difficult [4,5]. In recent years, numerous scholars have attempted to apply deep learning to groundwater burial depth prediction. Feed-forward neural networks were applied to predict groundwater levels in Chandpur district, Bangladesh by Husna et al. (2016) [6]. The results found that the artificial neural network (ANN) predicted groundwater levels within ten weeks with reasonable error; beyond ten weeks, the accuracy of the predictions rapidly decreased. A large number of studies have found that a single prediction model cannot extract the characteristics of groundwater burial depth changes well, cannot reduce the volatility of the series, and the mixing of high-frequency data with low-frequency data cannot be reduced leading to a general prediction effect. Some scholars have combined data processing models with predictive models for groundwater prediction. He et al. (2021) used the K-nearest neighbor (KNN) method to calculate water levels, and reconstructed the spatio-temporal dataset and the long and short-term memory model (LSTM) to construct a spatio-temporal KNN-LSTM prediction model for groundwater levels considering spatio-temporal factors [7]. The coupled model was found to have higher prediction accuracy than the single prediction model. Liu et al. (2021) proposed a deep learning model of long- and short-term memory recurrent neural networks combined with non-fully connected neural networks to improve the accuracy of groundwater burial depth prediction [8]. The results show that the coupled model reduces prediction errors by more than 80% compared to a single prediction model. The groundwater data were subjected to complementary ensemble empirical modal decomposition (CEEMD) and then predicted by a random forest model (RF), by Fu et al., (2020) [9]. The method reduces the frequency fluctuations of the groundwater burial depth time series and improves the prediction accuracy, with the proposed root mean square error of the prediction model reaching 0.03 m. Liang et al. (2020) constructed an EEMD-PSO-ELM groundwater burial depth prediction model by assembling a combination of empirical modal decomposition EEMD, particle swarm operation (PSO) and extreme learning machine (ELM) [10]. Model validation was carried out on the groundwater burial depth of Youyi Farm in Sanjiang Plain, Heilongjiang Province, and the study showed that both EEMD and PSO can effectively improve the prediction accuracy of the ELM neural network, and the ELM neural network has great application prospects in regional groundwater burial depth prediction. Most papers found that the decomposition-forecasting model is more conducive to the prediction of non-linear, non-stationary time series than the use of a single model for forecasting. However, there are still high frequency, unstable components in the eigenmodal component after primary modal decomposition. It can be noticed that there are large fluctuations in a few components of the primary decomposition, resulting in poor prediction of these components. In order to reduce the impact of the high-frequency eigenmodal function IMF1 generated by the ensemble empirical modal decomposition on the prediction results, Yin et al., (2020) used wavelet packet decomposition to further decompose the IMF1 subseries into several subseries to further improve the prediction accuracy of the model [11]. Wang et al. (2017) performed a variational modal decomposition VMD of the high-frequency component of the empirical modal decomposition of the complementary ensemble to predict air quality, and the model outperformed the results after the primary modal decomposition [12]. The two-stage modal decomposition model is mostly used for power and wind and is currently less studied in groundwater depth prediction [13,14]. A two-stage modal decomposition is carried out for the components, which fluctuate greatly after the primary decomposition and are not good at extracting features to further improve the prediction accuracy, and this method is currently less studied in groundwater burial depth prediction.

This paper proposes a groundwater burial depth prediction model, CEEMDAN-VMD-CNN-GRU, based on complete ensemble empirical mode decomposition with adaptive noise (CEEMDAN), variational modal decomposition (VMD), convolutional neural network (CNN), and gated recurrent unit (GRU). The model can effectively solve the influence of the high-frequency IMF components generated in the CEEMDAN decomposition on the prediction results. The high-frequency IMF components after the primary decomposition are further decomposed twice by VMD, and the final fully decomposed sequences are input into the CNN-GRU model, which can extract the implied features between the groundwater burial depth and the time subsequence and perform groundwater burial depth sequence prediction.

## 2. Models and Methods

### 2.1. CEEMDAN Model

In order to solve the problem of modal aliasing when the EMD algorithm decomposes the signal, the ensemble empirical mode decomposition (EEMD) algorithm adds uniformly distributed white noise to the signal to be decomposed to mitigate the modal aliasing phenomenon of the decomposed signal, however, there will be residual auxiliary noise, resulting in large reconstruction errors [15]. The complementary ensemble empirical mode decomposition (CEEMD) algorithm replaces the addition of uniformly distributed white noise with paired, mutually opposite white noise, eliminating residual auxiliary noise from the decomposed signal, but the essence remains unchanged; the decomposed signal still contains some residual auxiliary noise [16]. In order to solve the above problems, Torres et al. (2011) [17] proposed a new decomposition algorithm, complete ensemble empirical mode decomposition with adaptive noise (CEEMDAN), which not only solves the problem of residual noise transfer from high-frequency to low-frequency components during the decomposition of EEMD and CEEMD, but also eliminates the modal overlap caused by the decomposed signal [18,19], and improves the operational efficiency, accuracy, and completeness [20]. The breakdown process is as follows.

Define the operator $E_{j}(\cdot )$ as the j th IMF component of the signal after decomposition by EMD. White noise $w^{i}(t)$ is added to the original signal x(t) to become the new signal $x(t)+\varepsilon_{0}w^{i}(t)$, with $\varepsilon_{0}$ as the noise factor.

(1) The new signal $x(t)+\varepsilon_{0}w^{i}(t)$ is decomposed by the EMD algorithm and the resulting M IMF components are averaged to obtain the first order IMF components:(1)$$IMF_{1}(t)=\frac{1}{M}\sum_{i=1}^{M}IMF_{1}^{i}(t)={\bar{IMF}}_{1}(t)$$

(2) The residuals from the decomposition of the first order IMF component are:(2)$$r_{1}(t)=x(t)-IMF_{1}(t)$$

(3) The M-decomposition of the signal $r_{1}(t)+\varepsilon_{1}E_{1}(w^{i}(t))$ yields the second order IMF component (i=1⋯M):(3)$$IMF_{2}(t)=\frac{1}{M}\sum_{i=1}^{M}E_{1}(r_{1}(t)+\varepsilon_{1}E_{1}(w_{i}(t))$$

(4) For n=2⋯N, the nth order residuals are:(4)$$r_{n}(t)=r_{n-1}(t)-IMF_{n}(t)$$

(5) Calculate the k+1th order IMF component.
(5)$$IMF_{\left(k+1\right)}(t)=\frac{1}{M}\sum_{i=1}^{M}E_{k}(r_{k}(t)+\varepsilon_{k}E_{k}(w^{i}(t))$$

(6) Repeat Equations (4) and (5) until the residual signal satisfies the termination condition of the decomposition and K modal components are obtained, with the residuals expressed as:(6)$$R(t)=X(t)-\sum_{i=1}^{k}IMF_{i}(t)$$

(7) The final original signal x(t) can be decomposed as:(7)$$X(t)=R(t)+\sum_{i=1}^{k}IMF_{i}(t)$$

### 2.2. VMD Model

Dragomiretskiy and Zosso, (2014) [21] proposed the Variational mode decomposition (VMD) model in 2014. This method can effectively solve the problems of endpoint effects and modal component mixing in EMD methods, which can reduce the non-smoothness of time series with high complexity and strong non-linearity and is more robust with respect to sampling and noise [22,23]. The CEEMDAN decomposition produces high frequency, high amplitude IMF components, and a secondary modal decomposition of these high frequency, high amplitude IMF components is performed by VMD. The VMD constrained variational model is as follows.
(8)$$\frac{\min}{\left\{u_{k},\omega_{k}\right\}}\left\{\sum_{k}{\left\|\partial_{t}\left[\left(\delta (t)+\frac{j}{\pi t}\right)\cdot u_{k}(t)\right]e^{-j\omega_{k}t}\right\|}_{2}^{2}\right\}$$
(9)$$s.t.\sum_{k}u_{k}=f$$

### 2.3. CNN Principle Structure

Convolutional Neural Network (CNN) consists of a convolutional layer, a pooling layer, and a fully connected layer [24]. The convolution and pooling layers extract the most significant features in the data, and the fully connected layer interprets the features; the model effectively reduces the complexity of data reconstruction and improves the quality of data features and prediction accuracy [25]. The internal feature vectors of the data extracted by CNN are constructed into a time series form in preparation for the next step of GRU model input. The CNN structure is shown in Figure 1.

### 2.4. Subsection

The GRU contains a reset gate, which determines how the new input information is combined with the previous memory, and an update gate, which defines the amount of previous memory saved to the current time step. GRU is an improvement of the LSTM algorithm, simplifying its structure and reducing the number of parameters to significantly increase the training speed and overcome the LSTM long-term dependency problem. The GRU structure is shown in Figure 2.

### 2.5. CEEMDAN-VMD-CNN-GRU Model

Currently, process-driven models and data-driven models are widely used for time series forecasting. The CEEMDAN-VMD-CNN-GRU model developed in this paper belongs to one of the data-driven modelling approaches. Data-driven models effectively avoid the drawbacks of traditional process-driven models that require approximate process simulations, large amounts of unknown input data and model parameters, and have the advantage of being fast, with few input parameters and no need to consider the physical mechanisms of the intermediate processes, and only need to find the best mapping relationships between input and output variables. The model in this paper has a strong data processing and mapping function compared to existing models. It only requires the groundwater level data collected as the input vector to be scaled and then directly input into the model for learning and mapping to obtain the output vector. In terms of forecast results, the forecast scheme meets the requirements of the Specification for Hydrological Intelligence Forecasting (GB/T 22482-2008) and achieves satisfactory results, which can be applied to actual forecasting.

The overall flow chart of the prediction model built in this paper combining CEEMDAN, VMD, CNN and GRU is shown in Figure 3. The specific steps are as follows.

(1) Primary modal decomposition: The time series of groundwater burial depth in the irrigation area of the Lower Yellow River People’s Victory Canal was decomposed by CEEMDAN to obtain multiple IMF components and one residual term.

(2) Secondary modal decomposition: VMD decomposition of the high frequency non-smooth IMF components generated by the primary modal decomposition of CEEMDAN.

(3) Extraction of data features: The CNN’s convolutional kernel emphasizes “windows” in space, allowing it to take into account the effects of data before and after the current data when processing time-series data, so that local data features can be extracted, and finally the local information can be aggregated to obtain the overall information. The introduced CNN can effectively compensate for the fact that recurrent neural networks do not take into account the spatial up-and-down problem and perform hierarchical feature extraction on the input data.

A CNN framework consisting of 2 convolutional layers, 2 pooling layers and a fully connected layer is constructed. According to the characteristics of the load data, convolutional layer 1 and convolutional layer 2 are designed as one-dimensional convolution and the ReLU activation function is selected for activation. In order to retain more information about data fluctuations, the pooling method of pooling layer 1 and pooling layer 2 is selected as maximum pooling. After the convolutional and pooling layers, the original data are mapped to the hidden feature space, and the fully connected layer structure is built to transform the output and extract the feature vector, with the activation function Sigmoid selected for the fully connected layer.

(4) Groundwater burial depth prediction: The GRU learns the feature vectors extracted by the CNN. A single-layer GRU structure is built to fully learn the proposed features in order to capture their internal patterns of variation. The main parameters of all models are shown in Table 1.

### 2.6. Evaluation Indicators

In order to assess the superiority and accuracy of the model forecasts, the mean absolute error (MAE) and root mean square error (RMSE) are used as evaluation indicators, and the Nash–Sutcliffe efficiency coefficient (NSE) is used to express the forecasting accuracy of each component and the overall forecast results. NSE close to 0 means that the prediction result is close to the average of the measured value, and the overall result is credible, but the error is large; the closer the NSE is to 1, the higher the accuracy of the prediction model, the better prediction effect, and the higher the credibility of the model. The calculation formula is as follows:(10)$$MAE=\frac{1}{n}\sum_{i=1}^{n}\left|y_{c}(i)-y_{o}(i)\right|,i=1,2,\ldots ,n$$
(11)$$RMSE=\sqrt{\frac{1}{n}\sum_{i=1}^{n}{\left[y_{o}(i)-y_{c}(i)\right]}^{2}},i=1,2,\ldots ,n$$
(12)$$NSE=1-\frac{\sum_{i=1}^{n}{\left[y_{c}(i)-y_{0}(i)\right]}^{2}}{\sum_{i=1}^{n}{\left[y_{0}(i)-{\bar{y}}_{0}\right]}^{2}},i=1,2,\ldots ,n$$
where, $y_{c}(i)$ is the predicted value, $y_{0}(i)$ is the measured value and ${\bar{y}}_{0}$ is the mean of the measured values.

## 3. Case Studies

### 3.1. Data Sources

The People’s Victory Canal Irrigation District is one of the large irrigation districts built in the lower reaches of the Yellow River after the founding of New China, located on the northern bank of the Yellow River in Henan Province. The irrigation area extends from the Yellow River in the south to the Wei River in the north, with a total area of 1486.84 km^2^. This study uses groundwater burial data from January 1993 to February 2022 from the Xinxiang People’s Victory Canal Irrigation Administration for modelling purposes to predict the average groundwater burial depth in the irrigation area. The location map of the study area is shown in Figure 4.

CEEMDAN-VMD-CNN-GRU is a data-driven model with the model input data being groundwater levels from January 1993 to February 2022. The 350 months of groundwater time series data were decomposed into n components by CEEMDAN model decomposition, and the component with high fluctuation was decomposed by VMD. Finally, the two decomposed components were fed into the CNN-GRU prediction model separately, and the predicted individual components were aggregated to obtain the final ground water level prediction data. The specific process is as follows. The groundwater depths for January 1993–February 2022 are shown in Figure 5.

### 3.2. CEEMDAN Primary Modal Decomposition

The groundwater burial depth series of the People’s Victory Canal irrigation area for a total of 350 months from January 1993 to February 2022 were decomposed by CEEMDAN to reduce the complexity of the groundwater burial depth series and improve the prediction accuracy. Set the noise standard deviation ratio to 0.2, the number of noise additions to 300, and the maximum number of allowed sieving iterations to 500. The final breakdown is into seven IMF subscales and one residual term. The results of the primary modal decomposition are shown in Figure 6.

According to Figure 6, the decomposed components are ordered from high to low frequencies as IMF1-IMF7, with Res as the residual term. The remaining trends show that the depth of groundwater burial is on an increasing trend for the period January 1993–February 2022. The frequency of the IMF1~IMF3 components is relatively high and volatile, and the trend stability is not high, which greatly affects the subsequent prediction accuracy. For this purpose, the secondary modal decomposition of IMF1 to IMF3 is carried out by means of the VMD model.

### 3.3. VMD Secondary Modal Decomposition

The secondary modal decomposition of the IMF1 to IMF3 components after the primary modal decomposition is carried out by VMD, and the decomposition yields the VIMF1-VIMF3 components and the VRes residual term. The modal number is set to 4, the penalty factor to 1500 and the convergence tolerance tol to 2 × 10^−6^. The results of the secondary modal decomposition of the VMD are shown in Figure 7. The components of the secondary modal decomposition are smoother than the IMF1~IMF3 components of the primary modal decomposition, with much reduced frequency fluctuations and enhanced regularity, greatly improving the accuracy of subsequent groundwater depth prediction.

As shown in Figure 7, the high frequency components of CEEMDAN are again decomposed by the VMD model, and the IMF1-IMF3 components, which have large frequency fluctuations and poor stability, are re-decomposed into the four VIMF1-VRes components. Combined with the CEEMDAN decomposition of IMF4-Res, the final decomposition of the groundwater depth of burial in the irrigation area of the People’s Victory Canal into seven components and two residual terms. After the groundwater burial depth series are decomposed by CEEMDAN-VMD, the volatility and non-smoothness of the series are substantially reduced, and the prediction accuracy and reliability of the model are improved.

### 3.4. CNN-GRU Model Prediction

The seven components of the quadratic modal decomposition and the two residual terms were fed into the CNN-GRU model. From a total of 350 months of data from January 1993 to February 2022, 280 months of data from January 1993 to February 2016 were selected as the training sample and 70 months of data from March 2016 to February 2022 were selected as the test sample.

The activation function of the CNN-GRU model is ReLU, the loss function of the CNN-GRU hybrid model is mean squared error (MSE), and the adaptive moment estimation (Adam) optimization algorithm is selected to optimize the model parameters with a number of 600 iterations. The predicted results for each component are shown in Figure 8. All components were superimposed to obtain the final March 2016–February 2022 groundwater burial depth prediction for the People’s Victory Canal irrigation area, as shown in Figure 9.

As can be seen from Figure 8 and Table 2, the final prediction is the worst, with a MAE of 0.1829 m, due to the most pronounced frequency fluctuations and poor smoothness of the VIMF1 component after the quadratic modal decomposition. The VIMF2-VRes are predicted well by the secondary modal decomposition of CEEMDAN. Among the IMF4-IMF7 components, the IMF4 component is poorly predicted, with more significant errors at the sequence endpoints, while the other components are well predicted and the results are reliable. As can be seen from Figure 9, the CEEMDAN-VMD-CNN-GRU predictions are good, with the smallest errors between the predicted and measured values, most of which lie between ±0.2 m.

Table 2 shows that VIMF1 has the largest prediction error of all the components and residual terms, with MAE reaching 0.1829 m and RMSE reaching 0.2371 m. The IMF4 component is the next best, with an MAE of 0.176 m and an RMSE of 0.217 m. VIMF1 has a Nash efficiency coefficient of 0.0677 m, with all other components and residual terms reaching 0.99 or more with high prediction accuracy. All components and residual terms were summed to obtain the final predicted groundwater depth of burial for the People’s Victory Canal irrigation area, with the predicted groundwater depth of 0.1824 m MAE, 0.2363 m RMSE and 0.9429 NSE from March 2016 to February 2022 to the measured data. It shows that the CEEMDAN-VMD-CNN-GRU model has high accuracy, good prediction results, and high model confidence.

### 3.5. Comparative Analysis

In order to verify that the secondary mode decomposition can better improve the prediction accuracy, the CEEMDAN-GRU and CEEMDAN-CNN-GRU models with primary mode decomposition and single GRU prediction models were selected for comparison. The predicted effects of groundwater depth of burial under different model decompositions are shown in Table 3.

As can be seen from Table 3 and Figure 10, the CEEMDAN-VMD-CNN-GRU model with quadratic modal decomposition has less error and higher prediction accuracy compared to the CEEMDAN-GRU, CEEMDAN-CNN-GRU, and single GRU prediction models with primary modal decomposition. For the single prediction model GRU, the quadratic modal decomposition of the CEEMDAN-VMD-CNN-GRU model predicted sequences with 47.81% and 49.42% lower errors MAE and RMSE and 21.38% higher prediction accuracy. For the primary modal decomposition of CEEMDAN-GRU and CEEMDAN-CNN-GRU, the secondary modal decomposition of CEEMDAN-VMD-CNN-GRU model predicted sequences with 33.60% and 16.52% lower error MAE, 36.27% and 33.26% lower RMSE, and 9.70% and 4.93% higher prediction accuracy. Although VMD of the IMF1-IMF3 components of the primary modal decomposition still produces the high-frequency component VIMF1, but at the same time produces other smoother low and medium frequency components, it is the VIMF1-VRes from the secondary modal decomposition that reduces the prediction difficulty and improves the stability of the predicted sequence. The results show that secondary modal decomposition of high frequency, low stability sequences after primary modal decomposition can improve prediction accuracy, i.e., VMD is effective.

The consistency of the predicted data with the observed data is examined by means of the violin plot (b) in Figure 11 to further analyze the applicability and stability of the model. It can be seen that the prediction models remain largely consistent with the distribution of the observed data, but the GRU model is less similar to the distribution of the observed data compared to the other models. As shown in Figure 11, the Taylor diagram (a), the reference value in the diagram is the original observation and the closeness to the reference value determines the superiority of the model. The Taylor diagram is determined by three main indicators of accuracy: the correlation coefficient, the standard deviation and the root mean square error. It is clear that the CEEMDAN-VMD-CNN-GRU model works best, taking into account the correlation coefficient, the standard deviation, and the root mean square error, and the CEEMDAN-VMD-CNN-GRU model is the closest to the observed data.

## 4. Discussion

Prediction using CNN or GRU alone is less effective. The CEEMDAN-GRU model has 21.40% and 20.63% lower MAE and RMSE and 10.65% higher prediction accuracy than the single prediction model GRU. The “decomposition-prediction” model has significant advantages for dealing with uncertain, non-stationary and non-linear series data.S CNN-GRU can effectively extract the coupling relationship and temporal correlation implied by the time series and improve the prediction accuracy. the CEEMDAN-CNN-GRU model reduces the prediction result error indicators MAE and RMSE by 20.46% and 15.08% compared with the CEEMDAN-GRU model and improves the prediction accuracy by 4.55%.The CEEMDAN-VMD-CNN-GRU model has the lowest MAE and RMSE and 94% prediction accuracy compared to the other models, indicating that the model outperforms the other models in terms of prediction performance.The CEEMDAN-VMD-CNN-GRU model has the advantage of being computationally fast, with few input parameters and no need to consider the physical mechanisms of intermediate processes, requiring only the search for the best mapping relationship between input and output variables, offering more options for all types of hydrological forecasting.The model input data are the average groundwater depth of the People’s Victory Canal irrigation area, which does not take into account the spatial distribution of groundwater depth, which is a shortcoming and limitation of the model prediction and a direction for improvement in the future.

## 5. Conclusions

This paper presents a groundwater burial depth prediction model based on CEEMDAN-VMD two-stage modal decomposition and cascaded CNN-GRU deep learning. CEEMDAN was used to decompose the initial groundwater burial depth series data, using VMD to further decompose the high frequency IMF1-IMF3 into smoother, lower frequency components. It is found that the secondary modal decomposition can effectively solve the problem of high frequency unstable IMF generated after primary modal decomposition. In the CNN-GRU prediction model, CNN is used to extract the implicit features of the coupling relationship between groundwater burial depth and time, and the feature data are used as input to the GRU model for prediction to improve the prediction accuracy. CEEMDAN-VMD-CNN-GRU has excellent prediction performance and good application prospects.

## Figures and Tables

**Figure 1 ijerph-20-00345-f001:**
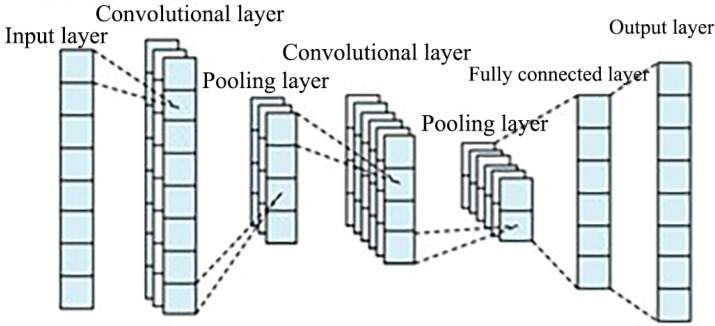
CNN structure diagram.

**Figure 2 ijerph-20-00345-f002:**
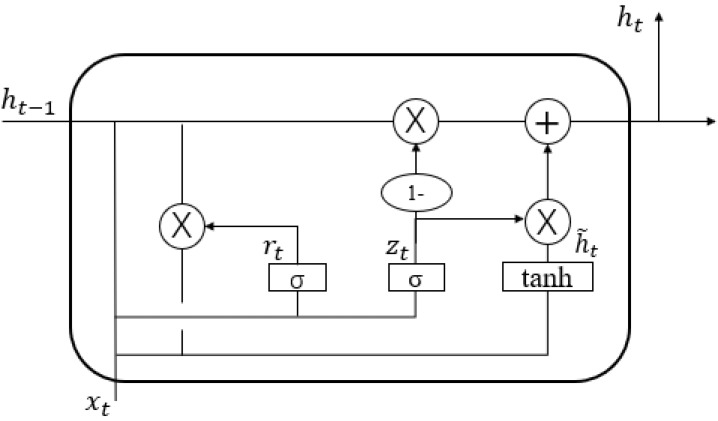
GRU structure.

**Figure 3 ijerph-20-00345-f003:**
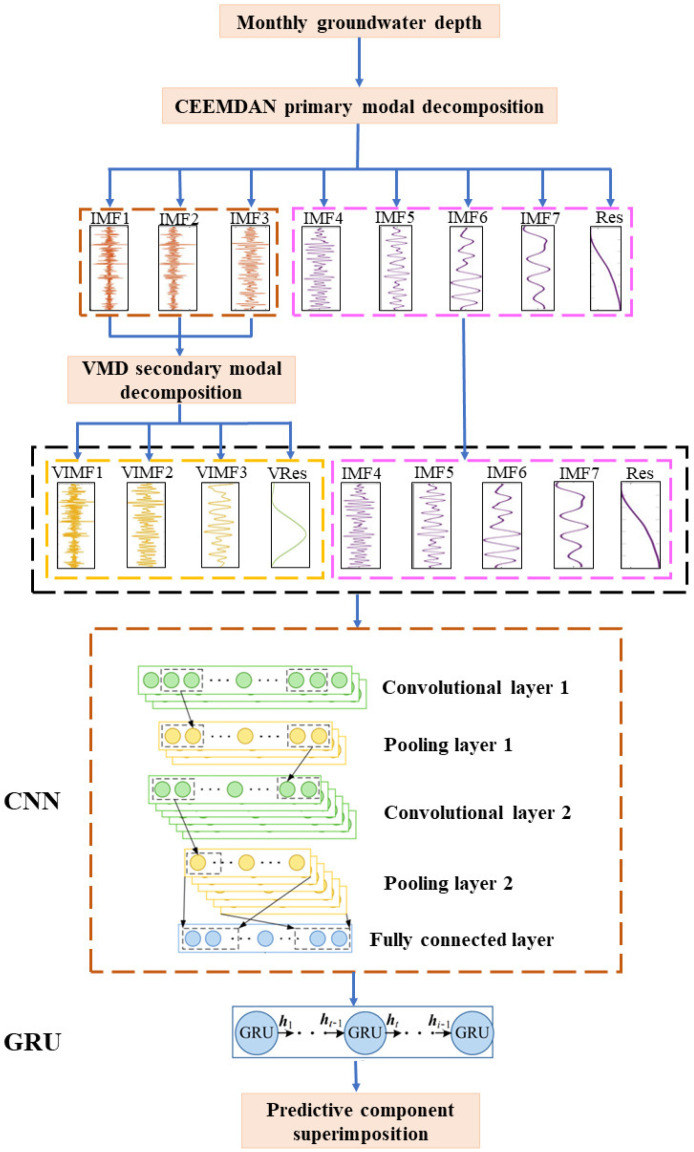
Groundwater depth prediction process.

**Figure 4 ijerph-20-00345-f004:**
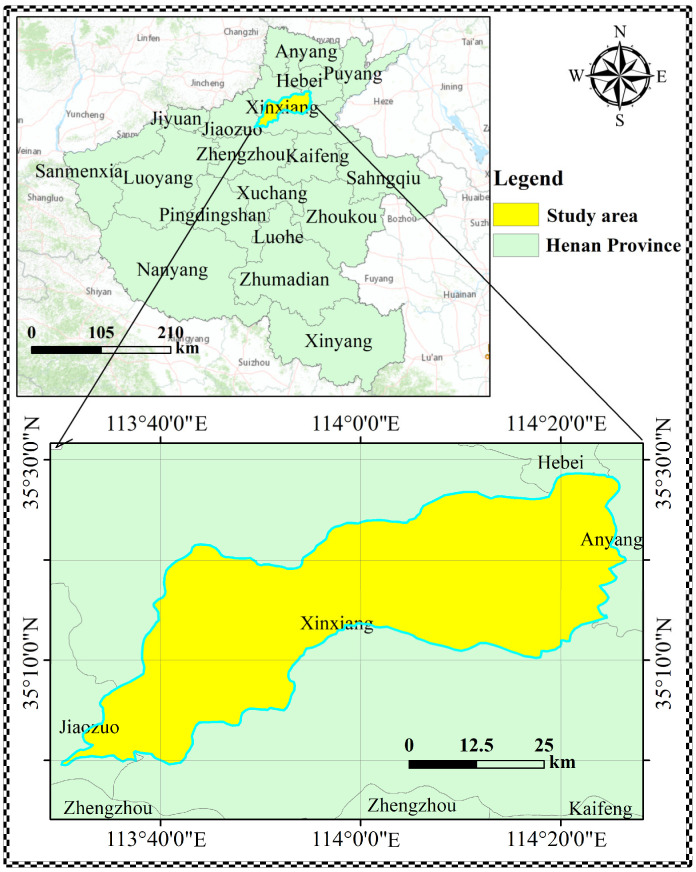
Location map of the study area.

**Figure 5 ijerph-20-00345-f005:**
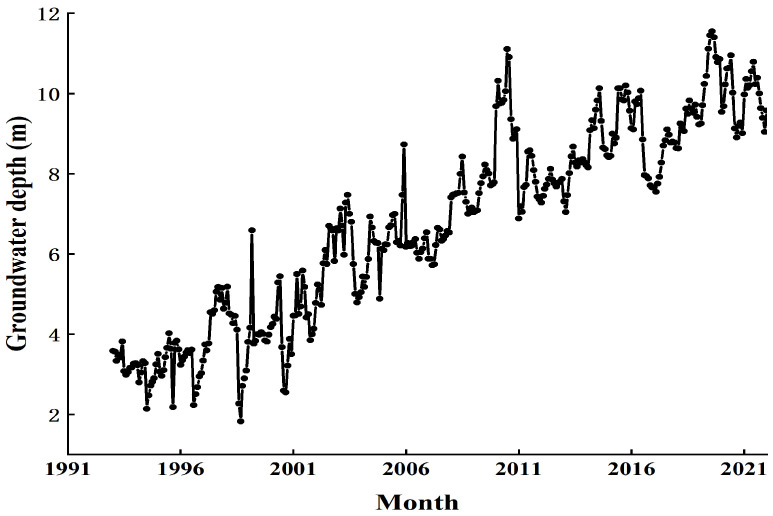
1993/1–2022/2 Groundwater depth.

**Figure 6 ijerph-20-00345-f006:**
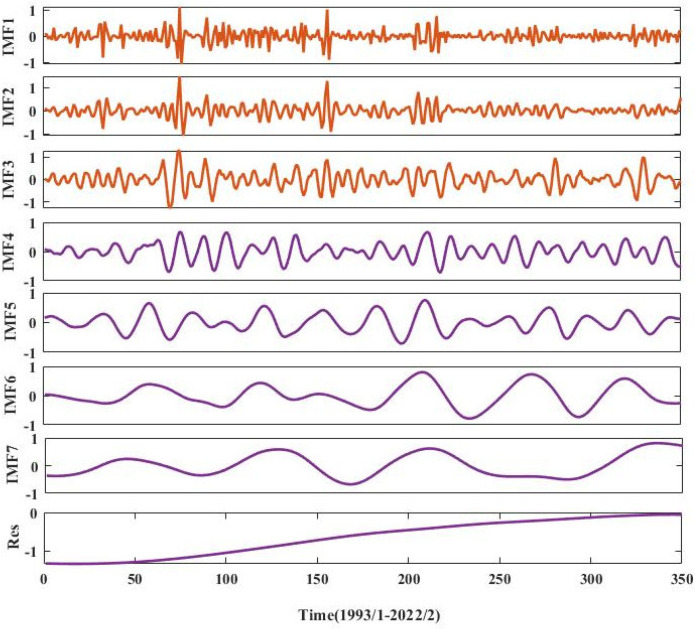
CEEMDAN primary modal decomposition results.

**Figure 7 ijerph-20-00345-f007:**
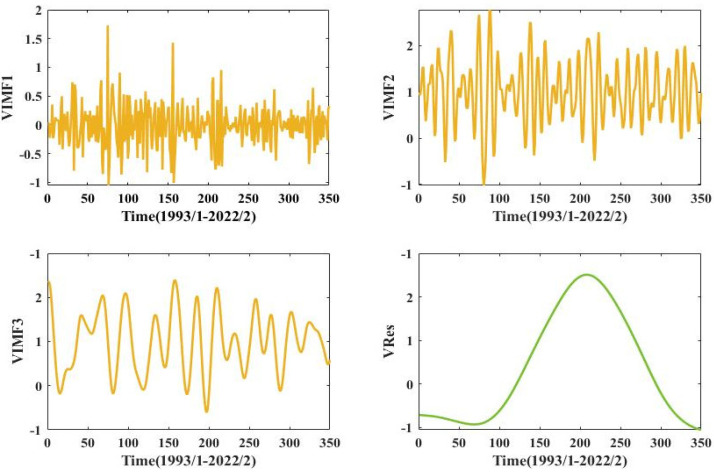
VMD decomposition results.

**Figure 8 ijerph-20-00345-f008:**
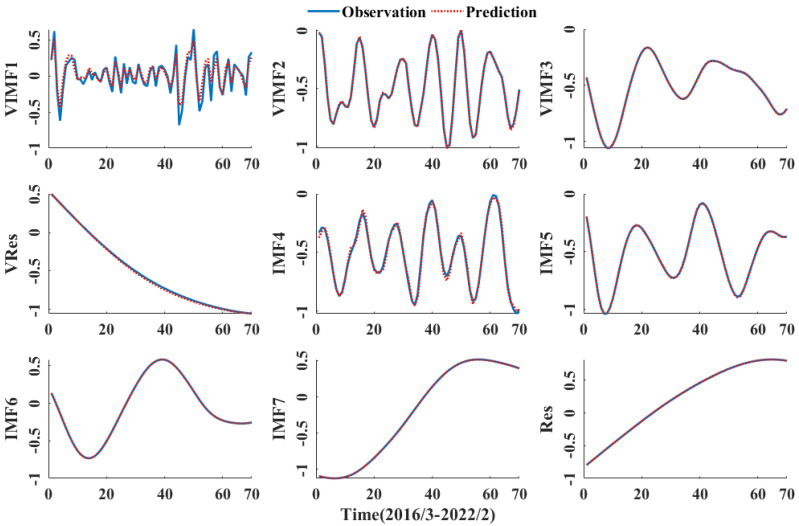
Predicted results for each component.

**Figure 9 ijerph-20-00345-f009:**
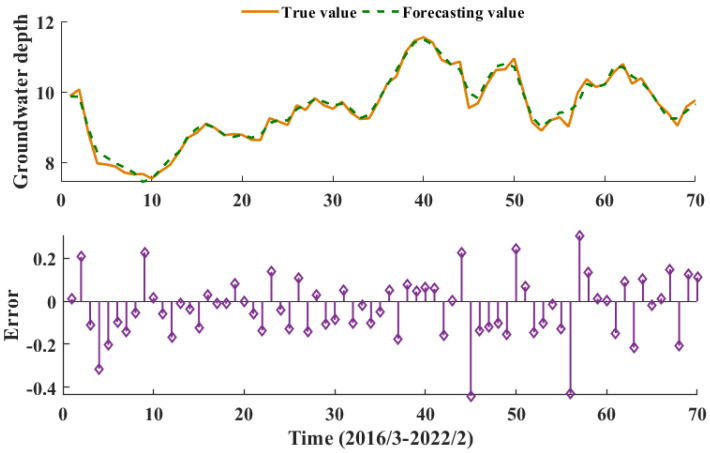
Predicted groundwater depth in the People’s Victory Canal irrigation area.

**Figure 10 ijerph-20-00345-f010:**
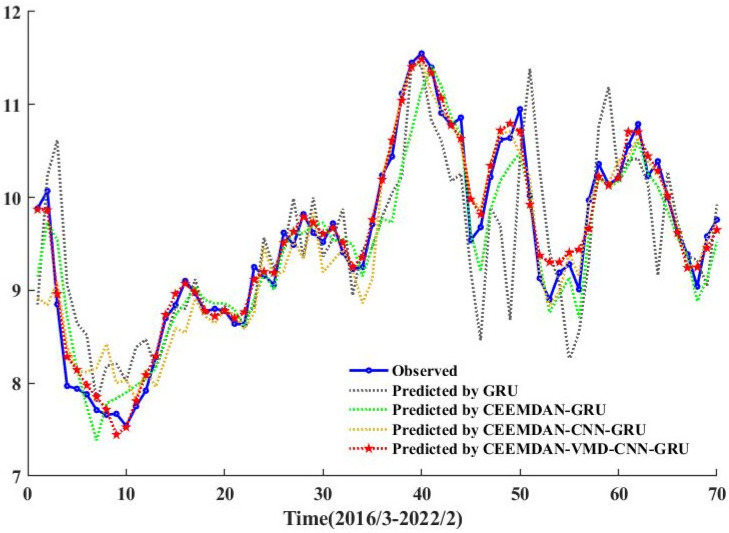
Comparison of prediction results from different models.

**Figure 11 ijerph-20-00345-f011:**
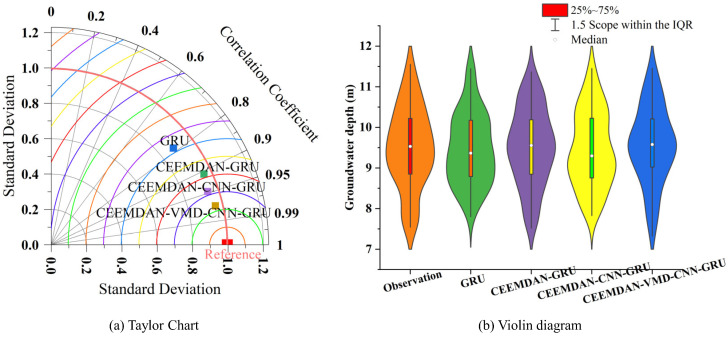
Comparison of the different models Taylor diagram (**a**) and violin diagram (**b**).

**Table 1 ijerph-20-00345-t001:** Main parameters of the model.

Layer	Parameter	Value
Input	Time step	1
CEEMDAN	Noise standard deviation ratio	0.2
	Number of noise additions	300
	Maximum number of allowed sieving iterations	500
VMD	Modal number	4
	Penalty Factor α	1500
	Convergence tolerances	2 × 10^−6^
CNN	Filters	32
	Kernel_size	32
	Padding	same
	Activation function	ReLU
GRU	Layers	5
	Number of neurons	{64, 64, 64, 64, 64}
	Optimisation algorithms	Adam
	Batch_size	20
	Epochs	80
	Error	MSE

**Table 2 ijerph-20-00345-t002:** Predicted effects of groundwater depth of burial for each component and overall.

Evaluation Indicators	MAE (m)	RMSE (m)	NSE
VIMF1	0.1829	0.2371	0.0677
VIMF2	0.0023	0.0029	0.9993
VIMF3	0.0001	0.0001	0.9999
VIMF4	0.0001	0.0001	0.9994
IMF4	0.0176	0.0217	0.9935
IMF5	0.0026	0.0032	0.9998
IMF6	0.0005	0.0007	0.9999
IMF7	0.0003	0.0004	1.0000
Residual	0.0005	0.0006	1.0000
Overall	0.1824	0.2363	0.9429

**Table 3 ijerph-20-00345-t003:** Calculated results of different model evaluation indicators.

Model	GRU	CEEMDAN-GRU	CEEMDAN-CNN-GRU	CEEMDAN-VMD-CNN-GRU
MAE	0.3495	0.2747	0.2185	0.1824
RMSE	0.4672	0.3708	0.3149	0.2363
NSE	0.7768	0.8595	0.8986	0.9429

## Data Availability

Data will be made available on request.

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
