# Peer review of "A Novel Groundwater Burial Depth Prediction Model Based on Two-Stage Modal Decomposition and Deep Learning"

_ijerph, 2022, doi:10.3390/ijerph20010345_

Round 1

Reviewer 1 Report

This article combines a signal decomposition technique with a neural network model to accurately forecast the groundwater burial depth in the studied area. To make a substantial contribution to the field, the paper should be improved in the following ways.

1.In Section 1, an overview of the previous studies should be provided, highlighting their flaws and weaknesses in order to introduce the paper's major topic.

2.In Section 2, the fundamental structure of the CEEMDAN-VMD-CNN-GRU model and its input data, essential parameters (modal number, convolution kernel, activation function, etc.) must be described in depth. The Figure 3 is incapable of illustrating the structure of the CNN model and the GRU model, as well as their integration process and how to utilize the model for prediction. Likewise, section 2.5 is too simple.

3. In the experimental sections, the modeling method of the prediction model, as well as model training and prediction process, should be detailed.

4. The discussion and analysis of the prediction results should be bolstered, including the interpretability of the model, the comparison with other prediction methods based on physical mechanisms, and the generalizability of the model, among other things.

5.Has the groundwater burial depth's spatial distribution pattern been taken into account? How should the spatial proximity effect of groundwater depth of burial be accounted for in the modeling process?

6.Verify the text on the final line of page 8.

7.It is recommended to include a map of the study area and some sample data to improve the paper's readability.

Reviewer 2 Report

The paper reports a new hybrid method for predicting ground water levels. Please consider these comments:

1) How do you train the CNN model?

2) The CNN model unknowns parameters such as filter size. Please add the values of these parameters to the paper?

3) Please compare the new model with orevious study?

4) is the model practical for other regions of the world?

5) Please explain the uncertainty resources of the modeling process such as inputs and model parameters.

6) what is your suggestion for the next papers?

Round 2

Reviewer 1 Report

The authors have done some work to improve their work. However, there are some important issues. Specifically, I am not completely convinced by the response 3. The Figure 3 should be redraw to illustrate the structure of the CNN model and the GRU model,  and how to integrated the CNN and GRU module?What's the meaning of "the two decomposed components were fed into the CNN-GRU prediction model separately"? What is the input data format for the convolution module, 1D sequence or 2D matrix?  What features are extracted by the CNN? How to aggregate the predicted individual components?

Reviewer 2 Report

The manuscript well revised based on the comments. 

Author Response

Thank you again for your work on the manuscript and I sincerely wish you good health and happiness every day.